# High-Intensity Ultrasound Processing Enhances the Bioactive Compounds, Antioxidant Capacity and Microbiological Quality of Melon (*Cucumis melo*) Juice

**DOI:** 10.3390/foods11172648

**Published:** 2022-08-31

**Authors:** Daniel Rodríguez-Rico, María de los Ángeles Sáenz-Esqueda, Jorge Armando Meza-Velázquez, Juan José Martínez-García, Jesús Josafath Quezada-Rivera, Mónica M. Umaña, Rafael Minjares-Fuentes

**Affiliations:** 1Facultad de Ciencias Químicas, Universidad Juárez del Estado de Durango, Gómez Palacio 35010, Mexico; 2Facultad de Ciencias Biológicas, Universidad Juárez del Estado de Durango, Gómez Palacio 35010, Mexico; 3Department of Chemistry, University of the Balearic Islands, 07122 Palma de Mallorca, Spain

**Keywords:** melon juice, high-intensity ultrasound, bioactive compounds, gallic acid, syringic acid, antioxidant capacity, microbiological quality

## Abstract

The bioactive compounds, antioxidant capacity and microbiological quality of melon juice processed by high-intensity ultrasound (HIUS) were studied. Melon juice was processed at two ultrasound intensities (27 and 52 W/cm^2^) for two different processing times (10 and 30 min) using two duty cycles (30 and 75%). Unprocessed juice was taken as a control. Total carotenoids and total phenolic compounds (TPC) were the bioactive compounds analyzed while the antioxidant capacity was determined by DPPH, ABTS and FRAP assays. The microbiological quality was tested by counting the aerobic and coliforms count as well as molds and yeasts. Total carotenoids increased by up to 42% while TPC decreased by 33% as a consequence of HIUS processing regarding control juice (carotenoids: 23 μg/g, TPC: 1.1 mg GAE/g), gallic acid and syringic acid being the only phenolic compounds identified. The antioxidant capacity of melon juice was enhanced by HIUS, achieving values of 45% and 20% of DPPH and ABTS inhibition, respectively, while >120 mg TE/100 g was determined by FRAP assay. Further, the microbial load of melon juice was significantly reduced by HIUS processing, coliforms and molds being the most sensitive. Thus, the HIUS could be an excellent alternative supportive the deep-processing of melon products.

## 1. Introduction

Melon (*Cucumis melo*) is one of the most consumed fruits worldwide as it contains naturally occurring vitamins (~60 mg/100 g [1]), minerals (~260 mg/100 g [1]), and pigments (>2000 μg/100 g [2]) that provide taste, health benefits, high antioxidant, and anti-inflammatory properties. Melon is rich in important vitamins and also a good source of pro-vitamin A [3]. However, its high level of water together with low acidity makes it a very perishable fruit resulting in high postharvest losses. In general, postharvest loss can be controlled by postharvest treatments, preservation technologies and deep processing technologies, melon juice being one of the most important deep processing products of melon with a high nutritional value [4].

Fruit juices are commonly subjected to thermal processing to assure their safety and extend their shelf-life. Nevertheless, thermal procedures, such as pasteurization or sterilization, lead to the loss of heat-sensitive nutrients, bioactive components and compromise the fresh taste [5]. In fact, it has been found that heat-treated melon juice exhibits a strong cooked off-flavor [4], limiting the development of melon products. All this, together with the high consumer demand for high-quality and fresh-like fruit juices, has been encouraging food industries to look for mild preservation techniques that minimize the negative impact of thermal treatments [6]. Thus, non-thermal techniques have been the focus of many investigations in order not only to preserve the nutrients, but also to improve the quality of food processed products. Concerning melon juice processing, non-thermal technologies, such as ozone [7,8], ultraviolet (UV-C) irradiation [6,9] and ultrasound (US) [10], have been studied in order to improve or maintain the quality of juice. However, ozone and UV-C irradiation demonstrated to be unsuitable to maintain the overall quality of melon juice since promotes the degradation of different biocompounds modifying the color of the juice [7,8], or are unable to reduce the intrinsic microbial load of the juice [6], affecting its stability during storage. On the contrary, US has been shown to be suitable for inactivating some degradative enzymes, such as polyphenol oxidase, peroxidase and ascorbate peroxidase, promoting better color retention of melon juice and high cloud stability during storage [10], although there is no evidence of possible alterations in the bioactive compounds and microbial quality of melon juice as a consequence of US.

Interestingly, US has shown a strong antimicrobial capacity against a spectrum of microorganisms and has been recognized as a forthcoming technique to meet the FDA requirement of safety for fruits and vegetable products [11]. Further, it has been reported the application of US technology to food processing may increase the bioavailability and/or bioaccesibility of different biocompounds [12], such as phenolic compounds [13,14], carotenoids [15,16,17,18], anthocyanins [13,19,20], flavonoids [15] among others [21], resulting the increase of the functional properties associated to these biocompounds, i.e., antioxidant capacity [13,21], antimicrobial activity [12], antidiabetic activity [15], anticancer activity [12,13], etc. Thus, US processing could not only improve the quality and safety but also extend the shelf life of different fruit juices [10,22]. The extending shelf-life of juices promoted by US technology has been attributed, on the one hand, to the inactivation of degradative enzymes, such as polyphenol oxidase [23] and native microorganisms [23,24,25] and, on the other hand, to the breakdown of the cell-walls increasing the release of bioactive compounds with antioxidant properties, into the juice [14,20,23], improving the quality of this.

Based on all this, US technology could be a promising technological alternative that is able to retain the bioactive compounds and functional properties of melon juice, inactivating its intrinsic microbial load. Therefore, the main aim of this study was to assess the physicochemical properties, bioactive compounds—specifically phenolic compounds and carotenoids—antioxidant activity and microbial quality of melon juice processed by high intensity ultrasound (HIUS).

## 2. Materials and Methods

### 2.1. Juice Preparation

Cantaloupe melon fruits (*Cucumis melo* L.) were obtained from Matamoros, Coahuila, Mexico (25°32′50.1″ N, 130°09′06.2″ W). Fruits of uniform size (15–20 cm diameter), without physical damage and free of microbial contamination, were selected, washed and disinfected using 200 ppm sodium hypochlorite solution. The melon juice was extracted using a commercial juice extractor (Turmix de México S.A. de C.V., Ocoyoacac, Mexico). The soluble solids were adjusted to 3 °Bx using distilled water as in Fonteles et al. [10].

### 2.2. High-Intensity Ultrasound (HIUS) Processing

The acoustic treatment of melon juice was performed using a Branson Sonifier SFX-550 (Branson Ultrasonics Corp, Danbury, CT, USA) equipped with a ½ inch tip-horn, operating at 550 W and 20 kHz. Approximately 200 g of melon juice was placed into a double jacket vessel (250 mL). The HIUS treatments were carried out using two different acoustic intensities (27 and 52 W/cm^2^) for two different processing time (10 and 15 min) with two US duty cycles (30 and 75%). The temperature of the samples was controlled by recirculating water at 10 ± 2 °C through the jacked vessel.

Melon juice was heat pasteurized for comparison purposes. Thus, around 200 g of melon juice was heat-treated at 65 °C for 30 min in a double jacket vessel. Unprocessed melon juice was taken as a control sample. All experiments were recorded in triplicate.

Finally, melon juice was freeze-dried in a FreeZone 6 Benchtop Lyophilizer (Labconco, Kansas City, MO, USA) operating at −40 °C and 0.07 mBar for posterior determination of total carotenoids, total phenolic compounds, individual phenolic compounds by HPLC-DAD, and antioxidant activity.

### 2.3. Centrifugal Sedimentation

The centrifugal sedimentation assay was carried out following the methodology previously described by Shen et al. [14] with slight modifications. Approximately 50 mL of melon juice were placed in a centrifuge tube and centrifuged at 3500× *g* for 15 min. The supernatant was discarded and the sediment was weighed. The centrifugal sedimentation rate was calculated as follows:(1)$$SR\%=\left(\frac{m_{1}}{m_{2}}\right)\times 100,$$
where *m*_1_ is the weight of the precipitate after centrifugation and *m*_2_ is the weight of the juice before centrifugation.

### 2.4. Color Difference

The color difference of HIUS melon juice was determined using a CR-300 colorimeter (Konica Minolta Sensing, Ink., Tokyo, Japan). The *L** (brightness), *a** (greenness/redness) and *b** (blueness/yellowness) values were measured, and the Δ*E** (total color difference) was calculated as follows (Equation (2)) [14]:(2)$$\Delta E=\sqrt{{\left(L_{0}^{*}-L^{*}\right)}^{2}+{\left(a_{0}^{*}-a^{*}\right)}^{2}+{\left(b_{0}^{*}-b^{*}\right)}^{2}},$$
where $L_{0}^{*}$, $a_{0}^{*}$ and $b_{0}^{*}$ are the values of the control sample, and $L^{*}$, $a^{*}$ and $b^{*}$ the measured values corresponding to a processed sample. All parameters were measured in triplicate for each treatment.

### 2.5. Analysis of Total Carotenoids

The determination of total carotenoids in melon juice was determined following the methodology described by Borghesi et al. [26] with some modifications. Approximately 200 mg of the freeze-dried sample of melon juice were mixed with 2.5 mL of methanol, 5 mL of trichloromethane and 2.5 mL of distilled water. The mixture was homogenized at 13,000 rpm for 1 min using an ultraturrax IKA T18D (IKA Works, Wilmington, NC, USA) and subsequently centrifuged at 4050× *g* for 10 min at 4 °C. The lipophilic phase was separated using a Pasteur pipette and the remainder was re-extracted three times using 5 mL of trichloromethane. All extracts were concentrated in a concentrator vacufuge^®^ plus (Eppendorf, São Paulo, Brazil) until 5 mL. The total carotenoids were measured in a spectrophotometer HACH at 465 nm. The concentration was determined using the equation (Equation (3)) [27].
(3)$$Total\;carotenoids=\frac{\left(ABS_{465}\right)\left(V\right)\left(10,000\right)}{\left(m\right)\left(\varepsilon \right)},$$
where *ABS*_465_ is the absorbance at 465 nm; *V* is the volume of solvent; *m* is the mass of the sample and *ε* is the molar absorbance of trichloromethane at 465 nm (2396 mol/L).

### 2.6. Total Phenolic Compounds

The total phenolic compounds were spectrophotometrically measured following the Folin–Ciocalteu method, using 96-well microplates, as previously described by Reyes-Avalos et al. [28]. Approximately 500 mg of lyophilized melon juice was suspended in 5 mL of methanol and continuously mixed at 4 °C for 24 h. The results were expressed as mg of gallic acid per g of dry matter (dm). All determinations were carried out in triplicate.

#### Identification of Individual Phenolic Compounds by HPLC-DAD

The phenolic compounds were extracted and analyzed by HPLC-DAD according to the method described by Reyes-Avalos et al. [28] with slight modifications. Approximately 500 mg of the freeze-dried melon juice were homogenized with 5 mL of methanol HPLC grade in an ultraturrax IKA T18D (IKA Works, Wilmington, NC, USA). The prepared samples were mixed at 4 °C for 24 h. Then, the samples were centrifuged at 4060× *g* for 20 min at 20 °C and the supernatant was filtered through a ∅ 0.45 μm PTFE filter before HPLC analysis. The chromatographic analysis was carried out using an HPLC Agilent 1200 (Agilent Technology, Palo Alto, Santa Clara, CA, USA) equipped with a diode array detector (DAD), a quaternary pump and two C_18_ 5-μm (250 mm × 4.6 mm) column connected in series. The temperature, flow rate, and injection loop were 20 °C, 0.5 mL/min y 20 μL, respectively. The mobile phase was comprised of (A) acetic acid 0.5% (B) methanol and (C) 80% acetonitrile. The mobile phase gradient was of 95% A, 2% B and 3% C at 2 min, 90% A, 4% B and 6% C at 8 min, 75% A, 10% B and 15% C at 15 min, 60% A, 10% B and 30% C at 30 min, and 95% A, 2% B and 3% C at 40 min. The different phenolic compounds were analyzed at three different wavelengths: 280, 316 and 365 nm. Thirteen individual phenolic standards (99%) were used to identify and quantify.

### 2.7. Antioxidant Activity

The samples used to determine the antioxidant activity were prepared as in total phenolic compounds. The prepared samples were continuously mixed at 4 °C for 24 h. A MultiSkan FC spectrophotometer and 96-wells plates were used for antioxidant activity determinations.

#### 2.7.1. Radical Scavenging by DPPH Assay

The effect of high-intensity ultrasound on the DPPH radical scavenging capacity (RSC) of the melon juice was measured as previously described by Ge et al. [29] with slight modifications. An aliquot of 10 μL of melon juice (100 mg/mL) was mixed with 190 μL of 2.5 mM DPPH solution in methanol. The mixture was incubated at 25 °C for 30 min and the decrease in absorbance was measured at 520 nm. The RSC was expressed as the percentage of DPPH radical inhibition and calculated as follows:(4)$$\mathrm{DPPH}\;\mathrm{Inhibition}\;\left(\%\right)=\left(\frac{A_{control}-A_{sample}}{A_{control}}\right)\times 100,$$
where *A_control_* and *A_sample_* refer to absorbance from control and sample, respectively.

#### 2.7.2. ABTS Free Radical Scavenging Assay

The effect of high-intensity ultrasound on the ABTS free radical scavenging capacity of melon juice was assessed as previously described Ge et al. [29] with slight modifications. An aliquot of 10 μL of melon juice (100 mg/mL) was mixed with 190 μL of 7 mM ABTS solution in methanol containing 2.5 mM potassium persulfate and incubated at 25 °C for 30 min. The change in the absorbance was recorded at 740 nm. The ABTS free radical scavenging was expressed as the percentage of ABTS free radical inhibition and calculated as follows:(5)$$\mathrm{ABTS}\;\mathrm{Inhibition}\;\left(\%\right)=\left(\frac{A_{control}-A_{sample}}{A_{control}}\right)\times 100,$$
where *A_control_* and *A_sample_* refer to absorbance from control and sample, respectively.

#### 2.7.3. FRAP Assay

The FRAP assay was carried out according to Hernández-Rodríguez et al. [30]. An aliquot of 10 μL of the sample was mixed with 190 μL of the FRAP solution and incubated at 25 °C for 30 min. The absorbance was measured at 593 nm. The antioxidant capacity measured by FRAP was expressed as milligrams of Trolox equivalent per gram of sample (mg TE/g).

### 2.8. Microbiological Quality

The microbial analysis of melon juice was performed according to the methodology of AOAC for aerobic (990.12), total coliforms (990.14) and yeast and mold (997.02) count plate. Proper serial dilutions were prepared by mixing sterilized distilled water followed by further decimal dilutions up to obtain colonies in the countable range (10–100 CFU/mL). Thus, 3M^TM^ Petrifilm^TM^ aerobic count plate, 3M^TM^ Petrifilm^TM^ coliform count plate and 3M^TM^ Petrifilm^TM^ yeast and mold count plate (3M Company, Saint Paul, MN, USA) were used. The plates were incubated at 35 ± 1 °C for 48 h for the aerobic count, at 35 ± 1 °C for 24 h for total coliforms and at 25 ± 1 °C for 5 days for yeast and molds. The yeast and molds were differentiated following the instructions of the 3M™ Petrifilm™ interpretation guide. Yeasts grow as small colonies with a defined edge, showing color ranges from pink-tan to blue-green color without a center focus (dark center), while molds grow as large colonies, with a diffuse edge and flat, exhibiting a wide variety of colors and center focus (dark colors). The number of colony-forming units (CFU) was expressed as logarithmic cycles (log (CFU/mL)).

### 2.9. Statistical Analysis

The results from centrifugal sedimentation, total carotenoids, total phenolic compounds, antioxidant capacity and microbiological inactivation were analyzed by one way ANOVA with a statistical significance level α = 0.05. The post-hoc analysis was performed using the Fisher’s least significant difference (LSD) test. All analytical determinations were performed in triplicate. All statistical analyses were performed in MINITAB software version 19 (Minitab Inc., State College, PA, USA).

## 3. Results and Discussion

### 3.1. Centrifugal Sedimentation

The effect of HIUS processing of melon juice on the percentage of centrifugal sedimentation is shown in Figure 1. The HIUS processing promoted a significant decrease in the centrifugal sedimentation of melon juice compared with pasteurization processing or control juice (*p* > 0.05). In particular, centrifugal sedimentation from those HIUS juices ranged from 12.1% to 14.2% while for pasteurized and control juices were 14.6% and 14.5%, respectively. For longer times of HIUS processing (30 min), the centrifugal sedimentation is more efficiently reduced when working at lower duty cycles (30%).

These results are concomitant with the study performed by Shen et al. [14], who evaluated the centrifugal sedimentation in apple juice treated with temperature-controlled ultrasound. These authors observed that the centrifugal precipitation of apple juice was reduced by around 35% with temperature-controlled ultrasound processing. Likewise, Rojas et al. [31] observed that the application of high-intensity ultrasound (>790 W/cm^2^) for long times (>6 min) avoided the pulp sedimentation in peach juice. The decreasing of centrifugal sedimentation in fruit juices treated with HIUS has been associated with the reduction of the particle size of fibrous material from juice, as a consequence of the shear forces generated by the high-intensity US [14,31]. Thus, the HIUS processing could improve the physicochemical stability of melon juice during storage by the reduction of the centrifugal sedimentation, extending its shelf-life.

### 3.2. Color Change

Color attributes are considered an important standard by which to evaluate the quality of fruit juice or related products since it affects the consumers’ acceptance [32]. Thus, the evaluation of possible color changes attributed to HIUS processing was carried out.

Figure 2 shows the results of color change (Δ*E*) of pasteurized and sonicated melon juice taking unprocessed melon juice as a reference. It can be seen that the processing, either pasteurization or sonication, promotes some difference in the color of the juices, denoting a higher Δ*E* in the sonicated samples. The Δ*E* ranged from 1.5 up to ~3 in HIUS processed juice, whereas in pasteurized juices this value was ~1.2. Interestingly, the lowest Δ*E* of sonicated juices (~1.5) was obtained in the juice processed at 27 W/cm^2^ for 10 min at 75% of duty cycle. Further, the color of this HIUS juice was similar to pasteurized juice (65°C—30 min) (*p* > 0.05). It is important to point out that color change from those juices treated at 27 W/cm^2^ for 10 min at 30% of duty cycle and 27 W/cm^2^ for 30 min at 75% of duty cycle cannot be seen by the naked eye since Δ*E* was lower than 2 [33] while observable color changes (Δ*E* ≥ 2) were obtained with the rest of HIUS conditions. The increase of Δ*E* by the effect of HIUS has also been observed in other fruit juices, such as kiwi juice [33] and apple juice [14], among others. Previously, Costa et al. [34] observed that high-intensity ultrasound promoted high color stability in pineapple juice which has been attributed to the lower polyphenol oxidase activity and the lower availability of oxygen in sonicated samples as a consequence of the liquid degasification process. Although the preservation of color characteristics as result of the inactivation of degradative enzymes by US has been widely reported [10,23,34], the minimal color changes observed in this study could also be associated with the release of antioxidant compounds into juice as a consequence of the breakdown of the cell walls caused by the cavitation phenomenon [35,36].

### 3.3. Total Carotenoids

The effect of HIUS processing of melon juice on total carotenoid content is shown in Figure 3. The results show a significant increase in the concentration of total carotenoids in processed samples, either pasteurization or ultrasound, compared with the control juice (23.6 μg/g). The concentration of total carotenoids in pasteurized juice was 26.3 μg/g, while in HIUS processed juices varied from 26–32.5 μg/g. The highest concentration was obtained at 27 W/cm^2^ for 10 min at 30% of duty cycle, whereas the lowest content was observed at 52 W/cm^2^ at 75% of duty cycle during 10 min.

Previously, Abid et al. [37] observed that carotenoids increased from ~1.22 up to ~1.55 μg/mL when apple juice was treated with ultrasound at 2 W/cm^2^ for 60 min. Recently, Ordóñez-Santos et al. [38] used ultrasound technology as an alternative treatment to Cape gooseberry (*Physalis peruviana* L.) juice processing. These authors observed that the concentration of carotenoids increased up to 90% when juice was treated at 210 W for 40 min.

It has been observed that carotenoid-rich juices exhibit high concentrations of carotenoids when they are processed using HIUS. This increase has been associated, on the one hand, with carotenoids’ migration into juice as a consequence of the cell wall disruption promoted by the cavitation phenomenon, and, on the second hand, with the inactivation of degradative enzymes responsible for the browning in fruit juices, such as polyphenol oxidase [23,37,39,40], as a result of HIUS application.

### 3.4. Total Phenolic Compounds (TPC)

Figure 4 shows the results of the content of TPC of the melon juice treated with HIUS. As can be seen, the ultrasound processing promoted the significant reduction of the TPC of melon juice (*p* < 0.05), being most noticeable at higher intensities (52 W/cm^2^). The TPC from the control juice was around 1.1 mg GAE/g dm while in sonicated juices varied from 0.75 up to 0.90 mg GAE/g dm. Interestingly, juices processed at 75% of duty cycle exhibited lower TPC than those juices processed at 30% of duty cycle. It is important to note that those juices processed at 27 W/cm^2^ at 30% of duty cycle exhibited similar TPC content as pasteurized juice (0.9 mg GAE/g dm) (*p* > 0.05).

The reduction of total phenolic compounds as a consequence of HIUS application has been previously observed by other authors [10,41,42]. For instance, Fonteles et al. [10] observed that the TPC from melon juice treated with US were reduced by between ~15% and ~35%. Likewise, Keenan et al. [42] reported that phenolic compounds from a fruit smoothie (banana, apple, strawberry and orange) underwent an important reduction (~25%) as a consequence of US processing. Overall, the degradation of phenolic compounds as a result of HIUS treatment has been associated, on the one hand, with the generation of free radicals, such as –OH by the cavitation phenomenon [43], and on the second hand, with the increase of temperature when long processing times (>10 min) are used [41]. In this study, the TPC decrease could be associated with the extended exposition of unbound polyphenols to acoustic energy since it has been observed that US processing facilities the release of bound phenolic compounds due to cell disruption, being more susceptible to acoustic degradation [20,23].

### 3.5. Identification of Individual Phenolic Compounds by HPLC-DAD

In order to gain more insight into the phenolic compounds present in melon juice, methanolic extracts were subjected to HPLC-DAD analysis. Two bioactive compounds were identified: gallic acid and syringic acid.

The effect of HIUS on the concentration of gallic acid (GA) and syringic acid (SA) in melon juice is shown in Figure 5. As can be seen, the HIUS promoted significant changes in the concentration of these bioactive compounds (*p* < 0.05). In particular, HIUS processing increased the GA content significantly (*p* < 0.05), reaching up to ~180 μg/g dm whereas pasteurization reduced it by ~25% regarding to control (~70 μg/g dm) (*p* < 0.05). Previously, Wang et al. [35] observed that GA content from strawberry juice increased from 0.12 mg/mL up to 0.29 mg/mL when juice was sonicated at 400 W for 16 min at 50% of duty cycle. These authors exposed that the increase in GA concentration might be the result of the attachment of hydroxyl radicals produced by US, to the aromatic ring [35] which could explain the increase of GA in this study. It is important to highlight that GA, one of the most important hydroxybenzoic acids and widely distributed in plants, has shown several biological activities in many diseases, including cardiovascular diseases, cancer, neurodegenerative disorders, and aging [44,45]. In fact, previous studies have revealed that GA exerts a protective role against kidney damage [46] and cerebral ischemic injury [47] due to its strong binding affinity on targeted noxious compounds which has been attributed to its antioxidant and anti-inflammatory activities [48].

On the other hand, SA was around 20 μg/g in control juice whereas in pasteurized juice it decreased to ~10 μg/g (*p* < 0.05). Interestingly, SA increased significantly as a consequence of HIUS processing (*p* < 0.05), ranging from ~27 up to ~60 μg/g. Noticeably, the juice processed at 27 W/cm^2^ for 30 min showed the lowest SA content, <35 μg/g of the samples treated with HIUS. On the other hand, in the other HIUS processed juices, SA content was higher than 40 μg/g (*p* < 0.05). Similar concentrations of SA have been previously observed in melon dehydrated, accounting for between ~21 μg/g in melon infrared-dehydrated, up to 52 μg/g when the melon was oven-dehydrated [3]. Thus, these results suggest the release of this type of bioactive compounds from inner cell into juice, confirming the rupture of cell walls by the shear forces generated by HIUS [35,36]. In the last years, SA (4-hydroxy-3,5-dimethoxybenzoic acid), a phenolic compound from the dimethoxybenzene subfamily of benzoic acids, has been the object of several studies mainly due to the functional properties related to this bioactive compound, such as strong antioxidant [49], anti-microbial [50] anti-osteoporotic [51], and anticancer [52,53] activities. In fact, recent studies have demonstrated that SA reduces cell proliferation, induces apoptosis, and alters autophagy by the modulation of the oxidative stress and DNA damage resulting in an effective therapeutic strategy in the treatment of several diseases, such as cancer or Alzheimer’s disease [52,53,54,55]. It is worthy to highlight that melon juice processed by HIUS could become considered a functional food due to bioactive compounds enrichment by the high concentrations of this phenolic acids and the carotenoids.

### 3.6. Antioxidant Activity of Melon Juice

The results of the effect of HIUS processing on the antioxidant capacity of melon juice are shown in Table 1. As can be seen, HIUS processing promoted a significant increase in the antioxidant capacity tested by DPPH, ABTS and FRAP assays (*p* < 0.05). In particular, HIUS processing increased the DPPH scavenging capacity of melon juice, reaching up to ~46% in juice processed at 27 W/cm^2^ for 30 min at 75% of duty cycle whereas in pasteurized and control juice was ~39 and ~29%, respectively. On the other hand, the ABTS free radical scavenging of melon juice processed by HIUS ranged from ~20% up to ~31% whereas in control and pasteurized juice was ~15% and ~9%, respectively (*p* < 0.05). Regarding to the antioxidant capacity by FRAP assay, the control and pasteurized juice exhibited values of ~40 and ~62, whereas in HIUS processed juices the antioxidant capacity reached values >110 mg TE/100 g dm (see Table 1).

The increase in the antioxidant capacity of fruit juices as a consequence of the sonication process has been widely reported by several authors [36,40,56]. Santhirasegaram et al. [40] observed the increase of the antioxidant capacity determined by DPPH, from 84.1% to 91.15%, and FRAP, from 360.71 up to 437.14 μg Ascorbic acid equivalent/mL, when mango juice was processed by ultrasound (<75 W) for 30 min. Likewise, Wang et al. [36] observed that ultrasound processing increased the antioxidant and radical scavenging capacities of kiwifruit juice. They observed that antioxidant activity by FRAP assay increased from 185.6 μmol/100 mL to 307.45 μmol/100 mL whereas the radical scavenging capacity by DPPH inhibition, was 1.6-fold higher than control (28.45%). Most of these authors [14,32] have attributed the increase of antioxidant capacity to the addition of a second hydroxyl group to the ortho- or para-positions to the aromatic ring of phenolic compounds during sonochemical reactions. Nevertheless, the enhancement of the antioxidant capacity observed in this study could be explained by the increase of individual phenolic compounds which have exhibited exceptionally high antioxidant activity, such as syringic acid (IC_50_ = 0.043 mM) [57] as well as its possible interaction with carotenoids [58]. It is important to point out that phenolic acids have become to exhibit better performance as free radical scavengers than antioxidant enzymes [57].

### 3.7. Microbiological Inactivation

Fruit juices have been considered a good alternative to increasing the consumption of fruits and vegetables [59]. Nevertheless, fruit juices are highly susceptible to contamination mainly by bacteria and some yeasts, either from contamination of fruit in the field or during its processing [60,61]. Microbial contamination can lead to the generation of off-flavors as well as render the products unsafe for direct consumption, due to the production of bio-toxins [62]. Thus, the effectiveness of HIUS processing to reduce the initial microbial load of melon juice was assessed.

The effect of HIUS processing on the aerobic bacterial count, coliform count, yeasts and molds count of melon juice is shown in Figure 6. As can be seen, the aerobic bacteria and coliforms accounted for around 3.88 log CFU/mL and 3.54 log CFU/mL, respectively, in the control juice. Interestingly, HIUS processing promoted a reduction between ~10 and 50% for aerobic bacteria (Figure 6a) and 100% for coliforms (Figure 6b), whereas the pasteurization showed the total inactivation for both cultures. Particularly, the highest inactivation of aerobic mesophiles by HIUS was obtained when melon juice was processed at 52 W/cm^2^ regardless duty cycle. On the other hand, yeasts were significantly reduced by HIUS processing, reaching a reduction between 12% and 20% comparing with the control juice (3.4 log CFU/mL) (*p* < 0.05) whereas the pasteurized juice exhibited a reduction around 38% (see Figure 6d).

Regarding molds count, HIUS processing promoted significant reduction (*p* < 0.05) while no significant difference was observed between control and pasteurized juice (2.8 log CFU/mL) (*p* > 0.05). Interestingly, molds count decreased to 2.0 log CFU/mL when melon juice was processed at 52 W/cm^2^ for 30 min and duty cycle of 30% (*p* < 0.05) (see Figure 6c).

It is important to mention that FDA has recognized ultrasound as a potential innovative technology for microbial inactivation in the fruit juices industry [11,63]. In fact, the reduction of the microbial load of fruit juices processed with US has been reported by several authors [24,25,40,62,64]. It has been observed that ultrasound was able to reduce below 2 log CFU/mL the natural microbiota of strawberry juice [24,25] whereas in mango juice, reduction of 26% and 100% aerobic mesophiles and coliforms, respectively, have been reported [40]. The effectiveness of ultrasound treatments depends, on the one hand, on the physicochemical and composition of the juice (°Bx, pH, titratable acidity), and on the other hand, on the resistance of microorganisms or the presence of bacteria or fungi spores [65]. In particular, fungi exhibit higher resistance compared with bacteria due to the cell wall composition [66]. Interestingly, the reduction observed by US processing has been attributed to the generation of mechanical shocks which lead to the destruction of the cell-wall yeasts by a depolymerization effect resulting in the lysis of cells and inactivation of certain enzymes [5].

Some authors have exposed that the lethal effect of ultrasound is increased when combined with other technologies, such as heat-treatment [24,25,39,64]. In fact, it has been reported that sonication at 20 °C reduces around 0.5 log CFU/mL in mesophiles, and up to 1 log CFU/mL in yeasts and molds whereas the complete inactivation of natural microbiota of apple juice was achieved when this was processed with US at 60 °C for 5 min (25 kHz and 70% power) [64]. Furthermore, Yildiz et al. [24,25] observed that ultrasound at mild temperature (55 °C for 3 min) kept below 2 log CFU/mL for 42 days at 4 °C the microbial load of strawberry juice as in the high hydrostatic pressure (300 MPa for 1 min) and thermal pasteurization (72 °C for 15 s) processing. However, the application of heat-treatments in the melon juice processing promotes the generation of cooked off-flavors components limiting the combination of US and heat-processing [4,67,68].

## 4. Conclusions

The physical stability, the bioactive compounds and microbial load of melon juice processed by HIUS were evaluated. Overall, the application of HIUS to the processing of melon juice improved the physical appearance, the content of bioactive compounds and the microbial quality of the juice. Thus, the HIUS enhanced the physical appearance of melon juice by reducing the sedimentation pulp, minimizing color change. Further, melon juice processed by HIUS was enriched with bioactive compounds by increasing the total carotenoids and the occurrence of gallic and syringic acids, augmenting the antioxidant capacity of the juice. It is important to highlight that melon juice processed by HIUS might be considered a functional food since gallic and syringic acids—two phenolic acids—have been involved in the prevention of diverse pathologies. Additional to this, HIUS treatment resulted in an effective procedure for the inactivation of microorganisms, specifically for coliforms and mold strains. These results demonstrate that HIUS might be a good technological alternative for the processing of thermal-sensitive fruit juices. Nevertheless, further studies are required to evaluate the effect of HIUS processing on the stability of the bioactive compounds of melon juice during storage and the sensorial characteristics.

## Figures and Tables

**Figure 1 foods-11-02648-f001:**
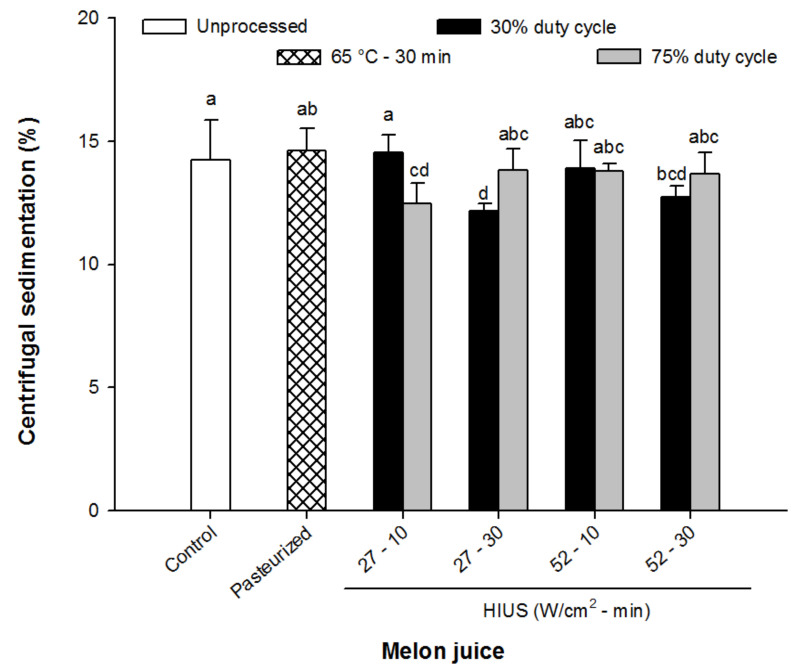
Centrifugal sedimentation of control (unprocessed), pasteurized and HIUS processed melon (*Cucumis melo*) juice. Different letters above the bars indicate statistical difference in treatments (*p* < 0.05) (*n* = 3).

**Figure 2 foods-11-02648-f002:**
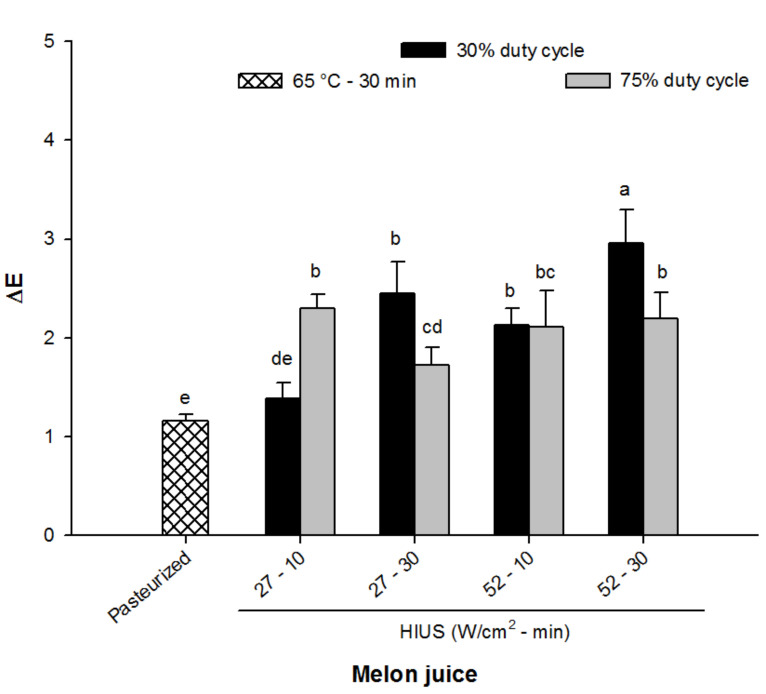
Color change of pasteurized and HIUS processed melon (*Cucumis melo*) juice. Different letters above the bars indicate statistical difference in treatments (*p* < 0.05) (*n* = 3).

**Figure 3 foods-11-02648-f003:**
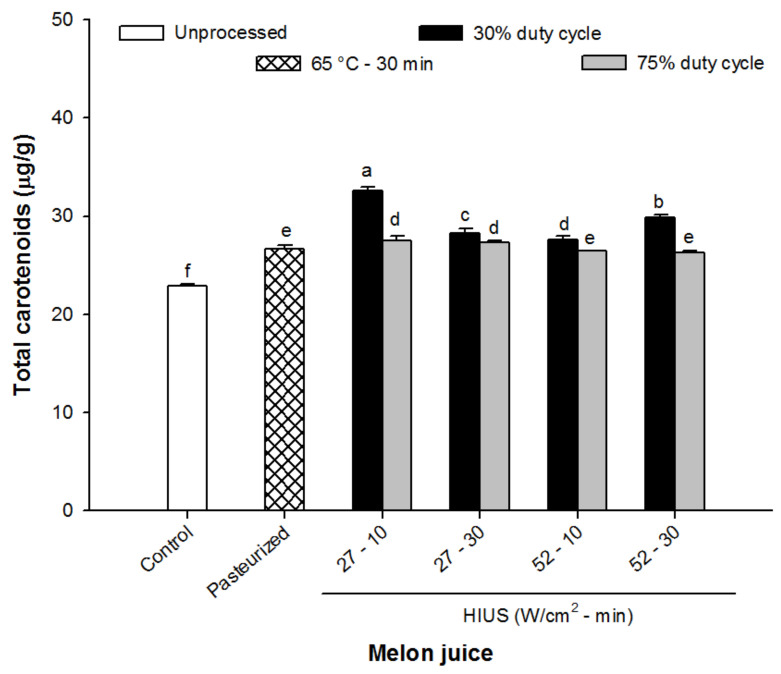
Total carotenoids content of control (unprocessed), pasteurized and HIUS processed melon (*Cucumis melo*) juice. Different letters above the bars indicate statistical difference in treatments (*p* < 0.05) (*n* = 3).

**Figure 4 foods-11-02648-f004:**
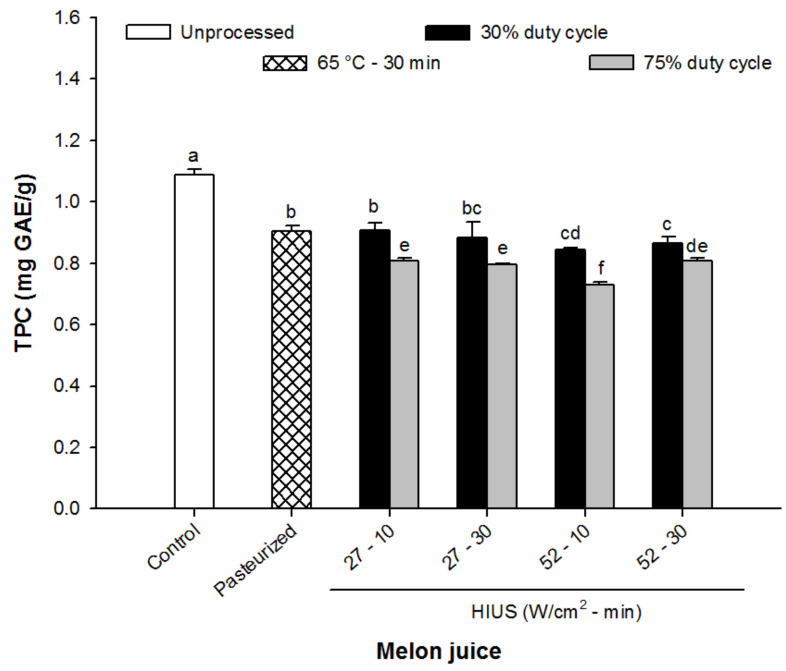
The total phenolic compound of control (unprocessed), pasteurized and HIUS processed melon (*Cucumis melo*) juice. Different letters above the bars indicate statistical difference in treatments (*p* < 0.05) (*n* = 3).

**Figure 5 foods-11-02648-f005:**
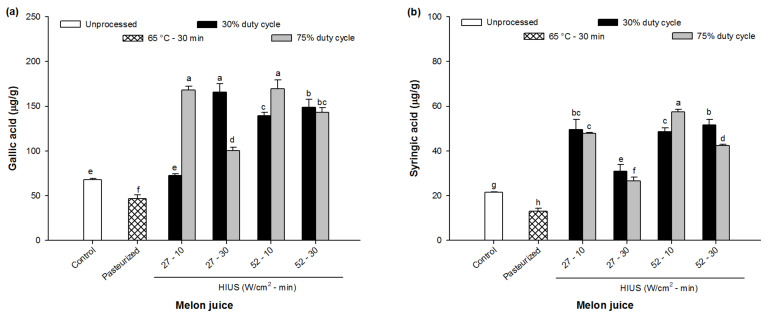
Individual phenolic compounds identified by HPLC-DAD in control (unprocessed), pasteurized and HIUS processed melon (*Cucumis melo*) juice: (**a**) gallic acid and (**b**) syringic acid. Different letters above the bars indicate statistical difference in treatments (*p* < 0.05) (*n* = 3).

**Figure 6 foods-11-02648-f006:**
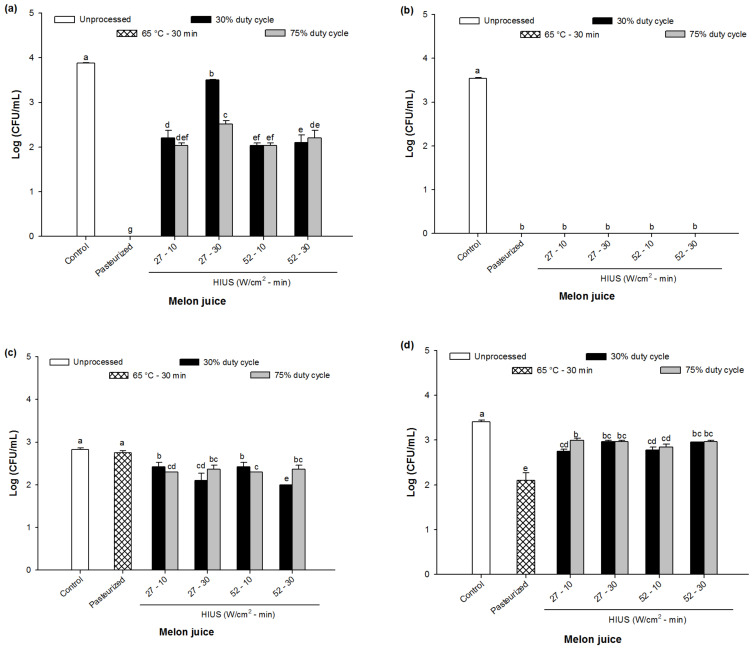
Microbial count of control (unprocessed), pasteurized and HIUS processed melon (*Cucumis melo*) juice: (**a**) aerobic bacterial count, (**b**) coliforms, (**c**) molds and (**d**) yeasts. Different letters above the bars indicate statistical difference in treatments (*p* < 0.05) (*n* = 3).

**Table 1 foods-11-02648-t001:** Antioxidant capacity of control (unprocessed), pasteurized and HIUS processed melon (*Cucumis melo*) juice. Different letters above the bars indicate statistical difference in treatments (*p* < 0.05) (*n* = 3).

Melon Juice			DPPH	ABTS	FRAP
			(%)	(%)	(mg TE/100 g dm)
*Control*			29.8 ± 0.9 ^g^	15.2 ± 1.3 ^e^	39.64 ± 1.5 ^f^
*Pasteurized*			39.3 ± 1.0 ^c^	8.65 ± 1.0 ^f^	62.29 ± 0.8 ^h^
** *HIUS* **					
**Intensity**	**Time**	**Duty Cycle**			
**(W/cm^2^)**	**(min)**	**(%)**			
27	10	30	33.1 ± 1.1 ^f^	26.0 ± 1.8 ^c^	89.79 ± 0.7 ^e^
		75	30.6 ± 1.6 ^g^	20.1 ± 0.7 ^d^	56.91 ± 1.3 ^g^
	30	30	39.5 ± 0.7 ^c^	28.0 ± 0.7 ^b^	111.7 ± 2.1 ^b^
		75	45.9 ± 1.0 ^a^	31.3 ± 0.1 ^a^	110.1 ± 1.9 ^bc^
52	10	30	41.8 ± 1.4 ^b^	30.5 ± 0.4 ^a^	108.7 ± 1.3 ^c^
		75	41.5 ± 1.6 ^b^	31.4 ± 0.4 ^a^	122.4 ± 3.3 ^a^
	30	30	37.5 ± 0.4 ^d^	30.9 ± 0.2 ^a^	91.01 ± 1.7 ^e^
		75	35.1 ± 0.4 ^e^	27.4 ± 1.0 ^bc^	95.02 ± 0.8 ^d^

## Data Availability

Data is contained within the article. The data used to support the findings of this study can be made available by the corresponding author upon request.

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
