# Peer review of "High-Intensity Ultrasound Processing Enhances the Bioactive Compounds, Antioxidant Capacity and Microbiological Quality of Melon (Cucumis melo) Juice"

_foods, 2022, doi:10.3390/foods11172648_

Round 1
Reviewer 1 Report
Recommendation: Minor Revision
The manuscript High-intensity ultrasound processing enhances the bioactive compounds, antioxidant capacity and microbiological quality of melon (Cucumis melo) juice, the methodology was reasonable and technically sound. Here are some issues to be addressed.
Minor concerns:
Comments to the Author:
The manuscript's title is appropriate.
Abstract: The Background of the abstract is well written. The main procedure and findings of the study are well expressed.
Introduction: A brief survey of existing literature, purpose, importance, and innovation of the research is well mentioned.
Tables and graphs are well prepared.
Point 1: In the last sentence of the abstract, it is recommended to give a suggestion for the future.
Point 2: In the introduction part of the article, it is important to mention the positive effects of ultrasound studies on enriching bioactive compounds. It is used in the enrichment of many different food products. I recommend the following articles
Erdal, Berna, Seydi Yıkmış, Nazan Tokatlı Demirok, Esra Bozgeyik, and Okan Levent. 2022. “Effects of Non-Thermal Treatment on Gilaburu Vinegar (Viburnum Opulus L.): Polyphenols, Amino Acid, Antimicrobial, and Anticancer Properties.” Biology 2022, Vol. 11, Page 926 11(6):926. doi: 10.3390/BIOLOGY11060926.
Yıkmış, Seydi, Filiz Aksu, Sema Sandıkçı Altunatmaz, and Başak Gökçe Çöl. 2021. “Ultrasound Processing of Vinegar: Modelling the Impact on Bioactives and Other Quality Factors.” Foods 10(8):1703. doi: 10.3390/FOODS10081703.
Yıkmış, Seydi, Berna Erdal, Esra Bozgeyik, Okan Levent, and Abdullah Yinanç. 2022. “Evaluation of Purple Onion Waste from the Perspective of Sustainability in Gastronomy: Ultrasound-Treated Vinegar.” International Journal of Gastronomy and Food Science 29:100574. doi: 10.1016/J.IJGFS.2022.100574.
Point 3: Line 67 Company information etc. of the commercial juicer should be given. Attention should be paid throughout the article.
Point 4: In the statistical part, the number of times the analyzes were repeated should be written. At the same time, the information of the minitab program should be written.
Point 5: for microbiological analysis; yeast and mold count was done in total as far as I understand. However, in figure 6, yeast and mold counts appear to be separate. I recommend that you explain this distinction in the method.
Author Response
Response to Reviewer 1 Comments
We really thank all the reviewer’s comments. We have done our best to attend all the reviewer’s suggestions.
Reviewer 1
Recommendation: Minor Revision
The manuscript High-intensity ultrasound processing enhances the bioactive compounds, antioxidant capacity and microbiological quality of melon (Cucumis melo) juice, the methodology was reasonable and technically sound. Here are some issues to be addressed.
Thanks for these comments, we really acknowledge all your suggestions
Minor concerns:
Comments to the Author:
The manuscript's title is appropriate.
Abstract: The Background of the abstract is well written. The main procedure and findings of the study are well expressed.
Introduction: A brief survey of existing literature, purpose, importance, and innovation of the research is well mentioned.
Tables and graphs are well prepared.
Point 1: In the last sentence of the abstract, it is recommended to give a suggestion for the future.
We really thank this suggest. The last phrase of abstract was modified as follow:
“Thus, the HIUS could be an excellent alternative supportive the deep-processing of melon products.”
Point 2: In the introduction part of the article, it is important to mention the positive effects of ultrasound studies on enriching bioactive compounds. It is used in the enrichment of many different food products. I recommend the following articles
Erdal, Berna, Seydi Yıkmış, Nazan Tokatlı Demirok, Esra Bozgeyik, and Okan Levent. 2022. “Effects of Non-Thermal Treatment on Gilaburu Vinegar (Viburnum Opulus L.): Polyphenols, Amino Acid, Antimicrobial, and Anticancer Properties.” Biology 2022, Vol. 11, Page 92611(6):926. doi: 10.3390/BIOLOGY11060926.
Yıkmış, Seydi, Filiz Aksu, Sema Sandıkçı Altunatmaz, and Başak Gökçe Çöl. 2021. “Ultrasound Processing of Vinegar: Modelling the Impact on Bioactives and Other Quality Factors.” Foods 10(8):1703. doi: 10.3390/FOODS10081703.
Yıkmış, Seydi, Berna Erdal, Esra Bozgeyik, Okan Levent, and Abdullah Yinanç. 2022. “Evaluation of Purple Onion Waste from the Perspective of Sustainability in Gastronomy: Ultrasound-Treated Vinegar.” International Journal of Gastronomy and Food Science29:100574. doi: 10.1016/J.IJGFS.2022.100574.
We really appreciate this suggestion. The different articles, including the above mentioned, were revised and the following paragraph was included in the introduction section.
Line 68-80
“Further, it has been reported the application of US technology to food processing may increase the bioavailability and/or bioaccesibility of different biocompounds [1], such as phenolic compounds [2,3], carotenoids [4-7], anthocyanins [2,8,9], flavonoids [4] among others [10], resulting the increase of the functional properties associated to these biocompounds, i.e. antioxidant capacity [2,10], antimicrobial activity [1], antidiabetic activity [4], anticancer activity [1,2], etc. Thus, US processing could not only improve the quality and safety but also extend the shelf life of different fruit juices [11,12]. The extending shelf-life of juices promoted by US technology has been attributed, on the one hand, to the inactivation of degradative enzymes, such as polyphenol oxidase [13] and native microorganisms [13-15], and, on the other hand, to the breakdown of the cell-walls increasing the release of bioactive compounds with antioxidant properties, into the juice [3,9,13], improving the quality of this.”
Point 3: Line 67 Company information etc. of the commercial juicer should be given. Attention should be paid throughout the article.
The company information of the commercial juicer was included as suggested
“… (Turmix de México S.A. de C.V., México ).”
Point 4: In the statistical part, the number of times the analyzes were repeated should be written. At the same time, the information of the minitab program should be written.
Thanks for this comment. The information about the replicates of analytical determinations as well as Minitab software information was included in the “Statistical analysis” section as follow.
Line 226-228
“All analytical determinations were performed in triplicate.”
“… MINITAB software version 19 (Minitab Inc., Pennsylvania, USA).”
Point 5: for microbiological analysis; yeast and mold count was done in total as far as I understand. However, in figure 6, yeast and mold counts appear to be separate. I recommend that you explain this distinction in the method.
Thanks for this comment. In fact, the count of yeast and molds was performed in the same petrifilm™ plate following the instructions guide; however, the colonies can be distinguish easily following the interpretation guide for microbiological test. After incubation period, we observed that molds colonies were noticeable lesser in melon juice processed by high-intensity ultrasound than in control juices. Based in this finding, it was decided to differentiate the yeast colonies from molds which was showed in figure 6.
In “2.7 microbiological quality” section, the following text was included (Line 216-222):
“The yeast and molds were differentiated following the instructions of the 3M™ Petrifilm™ interpretation guide. Yeasts grow as small colonies and defined edge, showing color ranges from pink-tan to blue-green color without center focus (dark center) while molds grow as large colonies, diffuse edge and flat, exhibiting wide variety of colors and center focus (dark colors).”
References
- Erdal, B.; Yıkmış, S.; Demirok, N.T.; Bozgeyik, E.; Levent, O. Effects of non-thermal treatment on gilaburu vinegar (viburnum opulus l.): Polyphenols, amino acid, antimicrobial, and anticancer properties. Biology 2022, 11, 926, https://doi.org/10.3390/biology11060926.
- Yıkmış, S.; Erdal, B.; Bozgeyik, E.; Levent, O.; Yinanç, A. Evaluation of purple onion waste from the perspective of sustainability in gastronomy: Ultrasound-treated vinegar. Int. J. Gastron. Food Sci. 2022, 29, 100574, https://doi.org/10.1016/j.ijgfs.2022.100574.
- Shen, Y.; Zhu, D.; Xi, P.; Cai, T.; Cao, X.; Liu, H.; Li, J. Effects of temperature-controlled ultrasound treatment on sensory properties, physical characteristics and antioxidant activity of cloudy apple juice. LWT 2021, 142, 111030, https://doi.org/10.1016/j.lwt.2021.111030.
- Yıkmış, S.; Aksu, F.; Altunatmaz, S.S.; Çöl, B.G. Ultrasound processing of vinegar: Modelling the impact on bioactives and other quality factors. Foods 2021, 10, 1703, https://doi.org/10.3390/foods10081703.
- Etzbach, L.; Stolle, R.; Anheuser, K.; Herdegen, V.; Schieber, A.; Weber, F. Impact of different pasteurization techniques and subsequent ultrasonication on the in vitro bioaccessibility of carotenoids in valencia orange (citrus sinensis (l.) osbeck) juice. Antioxidants 2020, 9, https://doi.org/10.3390/antiox9060534.
- Zhang, W.; Yu, Y.; Xie, F.; Gu, X.; Wu, J.; Wang, Z. High pressure homogenization versus ultrasound treatment of tomato juice: Effects on stability and in vitro bioaccessibility of carotenoids. LWT 2019, 116, 108597, https://doi.org/10.1016/j.lwt.2019.108597.
- Etzbach, L.; Pfeiffer, A.; Schieber, A.; Weber, F. Effects of thermal pasteurization and ultrasound treatment on the peroxidase activity, carotenoid composition, and physicochemical properties of goldenberry (physalis peruviana l.) puree. LWT 2019, 100, 69-74, https://doi.org/10.1016/j.lwt.2018.10.032.
- Sabino, L.B.d.S.; Filho, E.G.A.; Fernandes, F.A.N.; de Brito, E.S.; Júnior, I.J.d.S. Optimization of pressurized liquid extraction and ultrasound methods for recovery of anthocyanins present in jambolan fruit (syzygium cumini l.). Food Bioprod. Process. 2021, 127, 77-89, https://doi.org/10.1016/j.fbp.2021.02.012.
- Wu, Y.; Xu, L.; Liu, X.; Hasan, K.M.F.; Li, H.; Zhou, S.; Zhang, Q.; Zhou, Y. Effect of thermosonication treatment on blueberry juice quality: Total phenolics, flavonoids, anthocyanin, and antioxidant activity. LWT 2021, 150, 112021, https://doi.org/10.1016/j.lwt.2021.112021.
- Le, T.T.H.; Le, N.L. Antioxidant capacities and betacyanin lc-ms profile of red-fleshed dragon fruit juice (hylocereus polyrhizus) extracted by ultrasound-assisted enzymatic treatment and optimized by response surface methodology. J. Food Process. Preserv. 2021, 45, e15217, https://doi.org/10.1111/jfpp.15217.
- Roobab, U.; Aadil, R.M.; Madni, G.M.; Bekhit, A.E.-D. The impact of nonthermal technologies on the microbiological quality of juices: A review. Compr. Rev. Food Sci. Food Saf. 2018, 17, 437-457, https://doi.org/10.1111/1541-4337.12336.
- Fonteles, T.V.; Costa, M.G.M.; de Jesus, A.L.T.; de Miranda, M.R.A.; Fernandes, F.A.N.; Rodrigues, S. Power ultrasound processing of cantaloupe melon juice: Effects on quality parameters. Food Res. Int. 2012, 48, 41-48, http://dx.doi.org/10.1016/j.foodres.2012.02.013.
- Manzoor, M.F.; Xu, B.; Khan, S.; Shukat, R.; Ahmad, N.; Imran, M.; Rehman, A.; Karrar, E.; Aadil, R.M.; Korma, S.A. Impact of high-intensity thermosonication treatment on spinach juice: Bioactive compounds, rheological, microbial, and enzymatic activities. Ultrason. Sonochem. 2021, 78, 105740, https://doi.org/10.1016/j.ultsonch.2021.105740.
- Yildiz, S.; Pokhrel, P.R.; Unluturk, S.; Barbosa-Cánovas, G.V. Changes in quality characteristics of strawberry juice after equivalent high pressure, ultrasound, and pulsed electric fields processes. Food Eng. Rev. 2021, 13, 601-612, https://doi.org/10.1007/s12393-020-09250-z.
- Yildiz, S.; Pokhrel, P.R.; Unluturk, S.; Barbosa-Cánovas, G.V. Shelf life extension of strawberry juice by equivalent ultrasound, high pressure, and pulsed electric fields processes. Food Res. Int. 2021, 140, 110040, https://doi.org/10.1016/j.foodres.2020.110040.

Reviewer 2 Report
The submitted manuscript deals with the application of high-intensity ultrasound for pasteurization of melon juice. The paper highlights the benefits of high-intensity ultrasound processing over heat pasteurization for maintaining bioactive compounds in melon juice. The results are explained well citing similar researches. However there is still scope for elaborating the justification for results. The introduction needs to be improved explaining the importance of the study. I recommend this article for publication with following suggestions.
· The title of the manuscript looks like the result of the experiment. Kindly revise the title.
· Introduction
· Line 35: Mention the concentration of these compounds.
· Explain in short how ultrasound processing helps in preserving the quality of the juice.
· In the introduction, authors should cite relevant findings by other researchers justifying use of high intensity ultrasound for extending shelf life of juice and quality of processed juice.
· Hypothesis of the research needs to explain clearly.
· Materials and Methods
· Line 77: “Melon juice was pasteurized for comparison purposes.” Mention is as heat pasteurised as now cold pasteurization is also one of the pasteurization technique.
· Line 177: One way ANOVA or two way ANOVA?
· Results
· Standard deviation for each parameter is calculated but number of replications is not mentioned anywhere.
· Justification for results can be explained in more detail.
Author Response
Response to Reviewer 2 Comments
We really thank all the reviewer’s comments. We have done our best to attend all the reviewer’s suggestions.
Reviewer 2
The submitted manuscript deals with the application of high-intensity ultrasound for pasteurization of melon juice. The paper highlights the benefits of high-intensity ultrasound processing over heat pasteurization for maintaining bioactive compounds in melon juice. The results are explained well citing similar researches. However there is still scope for elaborating the justification for results. The introduction needs to be improved explaining the importance of the study. I recommend this article for publication with following suggestions.
Thanks for all your suggestions, we really thank these.
The title of the manuscript looks like the result of the experiment. Kindly revise the title.
Thanks for this comment. However, we write the title based on the guidelines described in Bowman and Kinnan [1], looking to have a strong title for high impact.
Introduction
Line 35: Mention the concentration of these compounds.
Regarding the concentration of vitamins, minerals and pigments, the concentration of these compounds was included in text as follow:
Line 35-36
“… vitamins (~60 mg/100g [2]), minerals (~260 mg/100g [2]), and pigments (>2000 mg/100 g [3]) …”
Explain in short how ultrasound processing helps in preserving the quality of the juice.
Thanks for this comment. The explanation about how US helps to preserve the quality of juice was included as follow:
Line 75-80
“The extending shelf-life of juices promoted by US technology has been attributed, on the one hand, to the inactivation of degradative enzymes, such as polyphenol oxidase [4] and native microorganisms [4-6], and, on the other hand, to the breakdown of the cell-walls increasing the release of bioactive compounds with antioxidant properties, into the juice [4,7,8], improving the quality of this.”
In the introduction, authors should cite relevant findings by other researchers justifying use of high intensity ultrasound for extending shelf life of juice and quality of processed juice.
Thanks for this comment. In order to give more information about the application of high intensity ultrasound for extending shelf-life and quality of processed juice, the next paragraph was included:
Line 68-80
“Further, it has been reported the application of US technology to food processing may increase the bioavailability and/or bioaccesibility of different biocompounds [9], such as phenolic compounds [7,10], carotenoids [11-14], anthocyanins [8,10,15], flavonoids [11] among others [16], resulting the increase of the functional properties associated to these biocompounds, i.e. antioxidant capacity [10,16], antimicrobial activity [9], antidiabetic activity [11], anticancer activity [9,10], etc. Thus, US processing could not only improve the quality and safety but also extend the shelf life of different fruit juices [17,18]. The extending shelf-life of juices promoted by US technology has been attributed, on the one hand, to the inactivation of degradative enzymes, such as polyphenol oxidase [4] and native microorganisms [4-6], and, on the other hand, to the breakdown of the cell-walls increasing the release of bioactive compounds with antioxidant properties, into the juice [4,7,8], improving the quality of this.”
Hypothesis of the research needs to explain clearly.
We really appreciate this suggestion. We included the next phrase in order to clarify the hypothesis of the research:
Line 81-86
“Based in all this, US technology could be a promising technological alternative able to retain the bioactive compounds and its functional properties of melon juice, inactivating its intrinsic microbial load.”
Materials and Methods
Line 77: “Melon juice was pasteurized for comparison purposes.” Mention is as heat pasteurised as now cold pasteurization is also one of the pasteurization technique.
Melon juice was heat pasteurized for comparative purposes. This was specified in the manuscript.
Line 177: One way ANOVA or two way ANOVA?
One way ANOVA was used for statistical analysis which has been specified in text.
Results
Standard deviation for each parameter is calculated but number of replications is not mentioned anywhere.
The number of replications has been included in Statistical analysis section (line 226-227). Additionally, this number was included in the title of the tables and figures.
Justification for results can be explained in more detail.
Thanks for this comment. Most of the beneficial of high intensity ultrasound have been attributed to cavitation phenomenon which causes temperature and pressure changes inside matrix or fluid, leading to reduction of the particle size and the cell disruption. In fruit juices, these effects promote the high stability of pulp due to homogeneous particle size and the release of cell components, mainly biocompounds, into juice by the rupture of cell walls [19-22]. Based on this, the following comments were included in order to give more detail about the justification of the results.
Line 271-275
“Although, the preservation of color characteristics as result of the inactivation of degradative enzymes by US has widely been reported [4,18,23], the minimal color changes observed in this study could also be associated to the release of antioxidant compounds into juice as consequence of the breakdown of the cell walls caused by cavitation phenomenon [24,25].”
Line 353-355
“Thus, these results suggest the release of this type of bioactive compounds from inner cell into juice, confirming the rupture of cell walls by the shear forces generated by HIUS [24,25].”
Line 363-365
“It is worthy to highlight that melon juice processed by HIUS could become considered a functional food due to bioactive compounds enrichment by the high concentrations of this phenolic acids and the carotenoids.”
References
- Bowman, D.; Kinnan, S. Creating effective titles for your scientific publications. VideoGIE 2018, 3, 260-261, 10.1016/j.vgie.2018.07.009.
- Eitenmiller, R.R.; Johnson, C.D.; Bryan, W.D.; Warren, D.B.; Gebhardt, S.E. Nutrient composition of cantaloupe and honeydew melons. J. Food Sci. 1985, 50, 136-138, https://doi.org/10.1111/j.1365-2621.1985.tb13294.x.
- Laur, L.M.; Tian, L. Provitamin a and vitamin c contents in selected california-grown cantaloupe and honeydew melons and imported melons. J. Food Comp. Anal. 2011, 24, 194-201, 10.1016/j.jfca.2010.07.009.
- Manzoor, M.F.; Xu, B.; Khan, S.; Shukat, R.; Ahmad, N.; Imran, M.; Rehman, A.; Karrar, E.; Aadil, R.M.; Korma, S.A. Impact of high-intensity thermosonication treatment on spinach juice: Bioactive compounds, rheological, microbial, and enzymatic activities. Ultrason. Sonochem. 2021, 78, 105740, https://doi.org/10.1016/j.ultsonch.2021.105740.
- Yildiz, S.; Pokhrel, P.R.; Unluturk, S.; Barbosa-Cánovas, G.V. Changes in quality characteristics of strawberry juice after equivalent high pressure, ultrasound, and pulsed electric fields processes. Food Eng. Rev. 2021, 13, 601-612, https://doi.org/10.1007/s12393-020-09250-z.
- Yildiz, S.; Pokhrel, P.R.; Unluturk, S.; Barbosa-Cánovas, G.V. Shelf life extension of strawberry juice by equivalent ultrasound, high pressure, and pulsed electric fields processes. Food Res. Int. 2021, 140, 110040, https://doi.org/10.1016/j.foodres.2020.110040.
- Shen, Y.; Zhu, D.; Xi, P.; Cai, T.; Cao, X.; Liu, H.; Li, J. Effects of temperature-controlled ultrasound treatment on sensory properties, physical characteristics and antioxidant activity of cloudy apple juice. LWT 2021, 142, 111030, https://doi.org/10.1016/j.lwt.2021.111030.
- Wu, Y.; Xu, L.; Liu, X.; Hasan, K.M.F.; Li, H.; Zhou, S.; Zhang, Q.; Zhou, Y. Effect of thermosonication treatment on blueberry juice quality: Total phenolics, flavonoids, anthocyanin, and antioxidant activity. LWT 2021, 150, 112021, https://doi.org/10.1016/j.lwt.2021.112021.
- Erdal, B.; Yıkmış, S.; Demirok, N.T.; Bozgeyik, E.; Levent, O. Effects of non-thermal treatment on gilaburu vinegar (viburnum opulus l.): Polyphenols, amino acid, antimicrobial, and anticancer properties. Biology 2022, 11, 926, https://doi.org/10.3390/biology11060926.
- Yıkmış, S.; Erdal, B.; Bozgeyik, E.; Levent, O.; Yinanç, A. Evaluation of purple onion waste from the perspective of sustainability in gastronomy: Ultrasound-treated vinegar. Int. J. Gastron. Food Sci. 2022, 29, 100574, https://doi.org/10.1016/j.ijgfs.2022.100574.
- Yıkmış, S.; Aksu, F.; Altunatmaz, S.S.; Çöl, B.G. Ultrasound processing of vinegar: Modelling the impact on bioactives and other quality factors. Foods 2021, 10, 1703, https://doi.org/10.3390/foods10081703.
- Etzbach, L.; Stolle, R.; Anheuser, K.; Herdegen, V.; Schieber, A.; Weber, F. Impact of different pasteurization techniques and subsequent ultrasonication on the in vitro bioaccessibility of carotenoids in valencia orange (citrus sinensis (l.) osbeck) juice. Antioxidants 2020, 9, https://doi.org/10.3390/antiox9060534.
- Zhang, W.; Yu, Y.; Xie, F.; Gu, X.; Wu, J.; Wang, Z. High pressure homogenization versus ultrasound treatment of tomato juice: Effects on stability and in vitro bioaccessibility of carotenoids. LWT 2019, 116, 108597, https://doi.org/10.1016/j.lwt.2019.108597.
- Etzbach, L.; Pfeiffer, A.; Schieber, A.; Weber, F. Effects of thermal pasteurization and ultrasound treatment on the peroxidase activity, carotenoid composition, and physicochemical properties of goldenberry (physalis peruviana l.) puree. LWT 2019, 100, 69-74, https://doi.org/10.1016/j.lwt.2018.10.032.
- Sabino, L.B.d.S.; Filho, E.G.A.; Fernandes, F.A.N.; de Brito, E.S.; Júnior, I.J.d.S. Optimization of pressurized liquid extraction and ultrasound methods for recovery of anthocyanins present in jambolan fruit (syzygium cumini l.). Food Bioprod. Process. 2021, 127, 77-89, https://doi.org/10.1016/j.fbp.2021.02.012.
- Le, T.T.H.; Le, N.L. Antioxidant capacities and betacyanin lc-ms profile of red-fleshed dragon fruit juice (hylocereus polyrhizus) extracted by ultrasound-assisted enzymatic treatment and optimized by response surface methodology. J. Food Process. Preserv. 2021, 45, e15217, https://doi.org/10.1111/jfpp.15217.
- Roobab, U.; Aadil, R.M.; Madni, G.M.; Bekhit, A.E.-D. The impact of nonthermal technologies on the microbiological quality of juices: A review. Compr. Rev. Food Sci. Food Saf. 2018, 17, 437-457, https://doi.org/10.1111/1541-4337.12336.
- Fonteles, T.V.; Costa, M.G.M.; de Jesus, A.L.T.; de Miranda, M.R.A.; Fernandes, F.A.N.; Rodrigues, S. Power ultrasound processing of cantaloupe melon juice: Effects on quality parameters. Food Res. Int. 2012, 48, 41-48, http://dx.doi.org/10.1016/j.foodres.2012.02.013.
- Toma, M.; Vinatoru, M.; Paniwnyk, L.; Mason, T.J. Investigation of the effects of ultrasound on vegetal tissues during solvent extraction. Ultrason. Sonochem. 2001, 8, 137-142, 10.1016/s1350-4177(00)00033-x.
- Vinatoru, M.; Toma, M.; Radu, O.; Filip, P.I.; Lazurca, D.; Mason, T.J. The use of ultrasound for the extraction of bioactive principles from plant materials. Ultrason. Sonochem. 1997, 4, 135-139, 10.1016/s1350-4177(97)83207-5.
- Dolas, R.; Saravanan, C.; Kaur, B.P. Emergence and era of ultrasonic’s in fruit juice preservation: A review. Ultrason. Sonochem. 2019, 58, 104609, https://doi.org/10.1016/j.ultsonch.2019.05.026.
- Ashokkumar, M. Applications of ultrasound in food and bioprocessing. Ultrason. Sonochem. 2015, 25, 17-23, http://dx.doi.org/10.1016/j.ultsonch.2014.08.012.
- Costa, M.G.M.; Fonteles, T.V.; de Jesus, A.L.T.; Almeida, F.D.L.; de Miranda, M.R.A.; Fernandes, F.A.N.; Rodrigues, S. High-intensity ultrasound processing of pineapple juice. Food Bioprocess Technol.2013, 6, 997-1006, https://doi.org/10.1007/s11947-011-0746-9.
- Wang, J.; Wang, J.; Ye, J.; Vanga, S.K.; Raghavan, V. Influence of high-intensity ultrasound on bioactive compounds of strawberry juice: Profiles of ascorbic acid, phenolics, antioxidant activity and microstructure. Food Control 2019, 96, 128-136, https://doi.org/10.1016/j.foodcont.2018.09.007.
- Wang, J.; Vanga, S.K.; Raghavan, V. High-intensity ultrasound processing of kiwifruit juice: Effects on the ascorbic acid, total phenolics, flavonoids and antioxidant capacity. LWT 2019, 107, 299-307, https://doi.org/10.1016/j.lwt.2019.03.024.

Reviewer 3 Report
These are my comments for this paper:
1) The introductory part is too superficially written; it is necessary to add a much larger number of references in order to emphasize the importance of the topic, novelty and better define the goal of the work itself
2) analysis of total carotenoids is questionable; why did you use only one absorbance to measure the total carotenoids? please, add references where the same methodology was used.
3) The paper does not reflect any novelty and further application possibilities; please, try to improve the whole paper in this stage of the review process.
Author Response
Response to Reviewer 3 Comments
We really thank all the reviewer’s comments. We have done our best to attend all the reviewer’s suggestions.
Reviewer 3
These are my comments for this paper:
We really acknowledge all your comments
1) The introductory part is too superficially written; it is necessary to add a much larger number of references in order to emphasize the importance of the topic, novelty and better define the goal of the work itself
Thanks for this comment, we really think this will improve the quality of the manuscript. Thus, introduction section was modifying following your suggestion, and the following paragraphs were included.
Line 55-65
“Concerning melon juice processing, non-thermal technologies, such as ozone [1,2], ultraviolet (UV-C) irradiation [3,4] and ultrasound (US) [5], have been studied in order to improve or maintain the quality of juice. However, ozone and UV-C irradiation demonstrated to be unsuitable to maintain the overall quality of melon juice since promotes the degradation of different biocompounds modifying the color of the juice [1,2], or are unable to reduce the intrinsic microbial load of the juice [3], affecting its stability during storage. On the contrary, US has showed to be suitable to inactive some degradative enzymes, such as polyphenol oxidase, peroxidase and ascorbate peroxidase, promoting better color retention of melon juice and high cloud stability during storage [5], although there is no evidence of possible alterations in the bioactive compounds and microbial quality of melon juice as consequence of US.”
Line 68-86
“Further, it has been reported the application of US technology to food processing may increase the bioavailability and/or bioaccesibility of different biocompounds [6], such as phenolic compounds [7,8], carotenoids [9-12], anthocyanins [7,13,14], flavonoids [9] among others [15], resulting the increase of the functional properties associated to these biocompounds, i.e. antioxidant capacity [7,15], antimicrobial activity [6], antidiabetic activity [9], anticancer activity [6,7], etc. Thus, US processing could not only improve the quality and safety but also extend the shelf life of different fruit juices [5,16]. The extending shelf-life of juices promoted by US technology has been attributed, on the one hand, to the inactivation of degradative enzymes, such as polyphenol oxidase [17] and native microorganisms [17-19], and, on the other hand, to the breakdown of the cell-walls increasing the release of bioactive compounds with antioxidant properties, into the juice [8,14,17], improving the quality of this.
Based in all this, US technology could be a promising technological alternative able to retain the bioactive compounds and its functional properties of melon juice, inactivating its intrinsic microbial load. Therefore, the main aim of this study was to assess the physicochemical properties, bioactive compounds, specifically phenolic compounds and carotenoids, its antioxidant activity and microbial quality of melon juice processed by high intensity ultrasound (HIUS).”
2) analysis of total carotenoids is questionable; why did you use only one absorbance to measure the total carotenoids? please, add references where the same methodology was used.
The spectrophotometric technique used to determine carotenoids is based in the measure of the absorbance at only one wavelength which can range from 377 to 510 nm, depending the type of carotenoid and the solvent used [20]. In this case, total carotenoids of melon juice were determined at 465 nm with an absorption coefficient of 2396 mol/L [20] since trichloromethane (chloroform) was used as a solvent. The determination of carotenoids by spectrophotometry has been widely reported by different authors [1,21-30] to determine different type of carotenoids from different sources.
3) The paper does not reflect any novelty and further application possibilities; please, try to improve the whole paper in this stage of the review process.
Thanks for this comment.
The novelty of the study focused in the application of high-intensity ultrasound as a technological alternative in the processing of thermal-sensitive fruit juices, such cantaloupe melon (Cucumis melo var. reticulatus). Melons, specifically cantaloupe melon, are highly valorized by its characteristic taste, the presence of different biocompounds such as B-carotene and vitamin C, and low caloric value. However, the high-water content together the low acidity of cantaloupe melon made of this a fruit highly perishable mainly by microorganism contamination which implies the application of different postharvest procedures to extend the fruit shelf-life, including the melon juice production. Concerning juice processing, thermal processing is the best technique to eliminate the pathogens microorganism assurance the safety of product. However, the heat processing of melon juice cause the loss of different bioactive compounds as well as the production of unpleasant aroma and flavor, limiting the deep-processing of melon products. Therefore, it is important to seek technological alternatives to promote the deep-processing of melon products, specifically melon juice, able to inactivate the intrinsic microbial load, minimizing the loss of biocompounds and its related functional properties. Within context, ultrasound could be a good alternative since it has demonstrated able to reduce or inactive foodborne microorganisms, preserving the biocompounds even some cases being increased.
Thus, in order to clarify the novelty of the study the introduction section was modified following your suggestion (Point 1).
Further, the next phrases were included in the “Results and discussion” section.
Line 249-251
“Thus, the HIUS processing could improve the physicochemical stability of melon juice during storage by the reduction of the centrifugal sedimentation, extending its shelf-life.”
Line 271-275
“Although, the preservation of color characteristics as result of the inactivation of degradative enzymes by US has widely been reported [5,17,31], the minimal color changes observed in this study could also be associated to the release of antioxidant compounds into juice as consequence of the breakdown of the cell walls caused by cavitation phenomenon [32,33].”
Line 353-355
“Thus, these results suggest the release of this type of bioactive compounds from inner cell into juice, confirming the rupture of cell walls by the shear forces generated by HIUS [32,33]. “
Line 363-365
“It is worthy to highlight that melon juice processed by HIUS could become considered a functional food due to bioactive compounds enrichment by the high concentrations of this phenolic acids and the carotenoids.”
In “Conclusions” section, the following phrases:
“These results demonstrate that HIUS might be a good technological alternative for the processing of thermal-sensitive fruit juices. Nevertheless, further studies are required to evaluate the effect of HIUS processing on the stability of the bioactive compounds of melon juice during storage and the sensorial characteristics.” (Line 466-469)
describe the potential application as well as the future studies.
References
- Fundo, J.F.; Miller, F.A.; Tremarin, A.; Garcia, E.; Brandão, T.R.S.; Silva, C.L.M. Quality assessment of cantaloupe melon juice under ozone processing. Innov. Food Sci. Emerg. Technol. 2018, 47, 461-466, https://doi.org/10.1016/j.ifset.2018.04.016.
- Sroy, S.; Fundo, J.F.; Miller, F.A.; Brandão, T.R.S.; Silva, C.L.M. Impact of ozone processing on microbiological, physicochemical, and bioactive characteristics of refrigerated stored cantaloupe melon juice. J. Food Process. Preserv. 2019, 43, e14276, https://doi.org/10.1111/jfpp.14276.
- Fundo, J.F.; Miller, F.A.; Mandro, G.F.; Tremarin, A.; Brandão, T.R.S.; Silva, C.L.M. Uv-c light processing of cantaloupe melon juice: Evaluation of the impact on microbiological, and some quality characteristics, during refrigerated storage. LWT 2019, 103, 247-252, https://doi.org/10.1016/j.lwt.2019.01.025.
- Kaya, Z.; Yıldız, S.; Ünlütürk, S. Effect of uv-c irradiation and heat treatment on the shelf life stability of a lemon–melon juice blend: Multivariate statistical approach. Innov. Food Sci. Emerg. Technol. 2015, 29, 230-239, https://doi.org/10.1016/j.ifset.2015.03.005.
- Fonteles, T.V.; Costa, M.G.M.; de Jesus, A.L.T.; de Miranda, M.R.A.; Fernandes, F.A.N.; Rodrigues, S. Power ultrasound processing of cantaloupe melon juice: Effects on quality parameters. Food Res. Int. 2012, 48, 41-48, http://dx.doi.org/10.1016/j.foodres.2012.02.013.
- Erdal, B.; Yıkmış, S.; Demirok, N.T.; Bozgeyik, E.; Levent, O. Effects of non-thermal treatment on gilaburu vinegar (viburnum opulus l.): Polyphenols, amino acid, antimicrobial, and anticancer properties. Biology 2022, 11, 926, https://doi.org/10.3390/biology11060926.
- Yıkmış, S.; Erdal, B.; Bozgeyik, E.; Levent, O.; Yinanç, A. Evaluation of purple onion waste from the perspective of sustainability in gastronomy: Ultrasound-treated vinegar. Int. J. Gastron. Food Sci. 2022, 29, 100574, https://doi.org/10.1016/j.ijgfs.2022.100574.
- Shen, Y.; Zhu, D.; Xi, P.; Cai, T.; Cao, X.; Liu, H.; Li, J. Effects of temperature-controlled ultrasound treatment on sensory properties, physical characteristics and antioxidant activity of cloudy apple juice. LWT 2021, 142, 111030, https://doi.org/10.1016/j.lwt.2021.111030.
- Yıkmış, S.; Aksu, F.; Altunatmaz, S.S.; Çöl, B.G. Ultrasound processing of vinegar: Modelling the impact on bioactives and other quality factors. Foods 2021, 10, 1703, https://doi.org/10.3390/foods10081703.
- Etzbach, L.; Stolle, R.; Anheuser, K.; Herdegen, V.; Schieber, A.; Weber, F. Impact of different pasteurization techniques and subsequent ultrasonication on the in vitro bioaccessibility of carotenoids in valencia orange (citrus sinensis (l.) osbeck) juice. Antioxidants 2020, 9, https://doi.org/10.3390/antiox9060534.
- Zhang, W.; Yu, Y.; Xie, F.; Gu, X.; Wu, J.; Wang, Z. High pressure homogenization versus ultrasound treatment of tomato juice: Effects on stability and in vitro bioaccessibility of carotenoids. LWT 2019, 116, 108597, https://doi.org/10.1016/j.lwt.2019.108597.
- Etzbach, L.; Pfeiffer, A.; Schieber, A.; Weber, F. Effects of thermal pasteurization and ultrasound treatment on the peroxidase activity, carotenoid composition, and physicochemical properties of goldenberry (physalis peruviana l.) puree. LWT 2019, 100, 69-74, https://doi.org/10.1016/j.lwt.2018.10.032.
- Sabino, L.B.d.S.; Filho, E.G.A.; Fernandes, F.A.N.; de Brito, E.S.; Júnior, I.J.d.S. Optimization of pressurized liquid extraction and ultrasound methods for recovery of anthocyanins present in jambolan fruit (syzygium cumini l.). Food Bioprod. Process. 2021, 127, 77-89, https://doi.org/10.1016/j.fbp.2021.02.012.
- Wu, Y.; Xu, L.; Liu, X.; Hasan, K.M.F.; Li, H.; Zhou, S.; Zhang, Q.; Zhou, Y. Effect of thermosonication treatment on blueberry juice quality: Total phenolics, flavonoids, anthocyanin, and antioxidant activity. LWT 2021, 150, 112021, https://doi.org/10.1016/j.lwt.2021.112021.
- Le, T.T.H.; Le, N.L. Antioxidant capacities and betacyanin lc-ms profile of red-fleshed dragon fruit juice (hylocereus polyrhizus) extracted by ultrasound-assisted enzymatic treatment and optimized by response surface methodology. J. Food Process. Preserv. 2021, 45, e15217, https://doi.org/10.1111/jfpp.15217.
- Roobab, U.; Aadil, R.M.; Madni, G.M.; Bekhit, A.E.-D. The impact of nonthermal technologies on the microbiological quality of juices: A review. Compr. Rev. Food Sci. Food Saf. 2018, 17, 437-457, https://doi.org/10.1111/1541-4337.12336.
- Manzoor, M.F.; Xu, B.; Khan, S.; Shukat, R.; Ahmad, N.; Imran, M.; Rehman, A.; Karrar, E.; Aadil, R.M.; Korma, S.A. Impact of high-intensity thermosonication treatment on spinach juice: Bioactive compounds, rheological, microbial, and enzymatic activities. Ultrason. Sonochem. 2021, 78, 105740, https://doi.org/10.1016/j.ultsonch.2021.105740.
- Yildiz, S.; Pokhrel, P.R.; Unluturk, S.; Barbosa-Cánovas, G.V. Changes in quality characteristics of strawberry juice after equivalent high pressure, ultrasound, and pulsed electric fields processes. Food Eng. Rev. 2021, 13, 601-612, https://doi.org/10.1007/s12393-020-09250-z.
- Yildiz, S.; Pokhrel, P.R.; Unluturk, S.; Barbosa-Cánovas, G.V. Shelf life extension of strawberry juice by equivalent ultrasound, high pressure, and pulsed electric fields processes. Food Res. Int. 2021, 140, 110040, https://doi.org/10.1016/j.foodres.2020.110040.
- Rodriguez-Amaya, D.B. A guide to carotenoid analysis in foods. 2001.
- Kumar, A.; Aggarwal, P.; Kumar, V.; Babbar, N.; Kaur, S. Melon-based smoothies: Process optimization and effect of processing and preservation on the quality attributes. Journal of Food Measurement and Characterization 2022, https://doi.org/10.1007/s11694-022-01466-3.
- Fundo, J.F.; Miller, F.A.; Garcia, E.; Santos, J.R.; Silva, C.L.M.; Brandão, T.R.S. Physicochemical characteristics, bioactive compounds and antioxidant activity in juice, pulp, peel and seeds of cantaloupe melon. Journal of Food Measurement and Characterization 2018, 12, 292-300, https://doi.org/10.1007/s11694-017-9640-0.
- Aadil, R.M.; Zeng, X.-A.; Han, Z.; Sahar, A.; Khalil, A.A.; Rahman, U.U.; Khan, M.; Mehmood, T. Combined effects of pulsed electric field and ultrasound on bioactive compounds and microbial quality of grapefruit juice. J. Food Process. Preserv. 2018, 42, e13507, https://doi.org/10.1111/jfpp.13507.
- Mallek-Ayadi, S.; Bahloul, N.; Baklouti, S.; Kechaou, N. Bioactive compounds from cucumis melo l. Fruits as potential nutraceutical food ingredients and juice processing using membrane technology. Food Science & Nutrition in Press, in Press, https://doi.org/10.1002/fsn3.2888.
- Medeiros, A.K.d.O.C.; Gomes, C.d.C.; Amaral, M.L.Q.d.A.; Medeiros, L.D.G.d.; Medeiros, I.; Porto, D.L.; Aragão, C.F.S.; Maciel, B.L.L.; Morais, A.H.d.A.; Passos, T.S. Nanoencapsulation improved water solubility and color stability of carotenoids extracted from cantaloupe melon (cucumis melo l.). Food Chem. 2019, 270, 562-572, https://doi.org/10.1016/j.foodchem.2018.07.099.
- de Oliveira, G.L.R.; Medeiros, I.; Nascimento, S.S.d.C.; Viana, R.L.S.; Porto, D.L.; Rocha, H.A.O.; Aragão, C.F.S.; Maciel, B.L.L.; de Assis, C.F.; Morais, A.H.d.A., et al. Antioxidant stability enhancement of carotenoid rich-extract from cantaloupe melon (cucumis melo l.) nanoencapsulated in gelatin under different storage conditions. Food Chem. 2021, 348, 129055, https://doi.org/10.1016/j.foodchem.2021.129055.
- Adiamo, O.Q.; Ghafoor, K.; Al-Juhaimi, F.; Mohamed Ahmed, I.A.; Babiker, E.E. Effects of thermosonication and orange by-products extracts on quality attributes of carrot (daucus carota) juice during storage. International Journal of Food Science & Technology 2017, 52, 2115-2125, https://doi.org/10.1111/ijfs.13490.
- Ahmed, I.A.M.; Al Juhaimi, F.; Özcan, M.M.; Uslu, N.; Babiker, E.E.; Ghafoor, K.; Osman, M.A.; Salih, H.A.A. A comparative study of bioactive compounds, antioxidant activity and phenolic compounds of melon (cucumis melo l.) slices dehydrated by oven, microwave and infrared systems. J. Food Process. Preserv. 2021, 45, e15605, https://doi.org/10.1111/jfpp.15605.
- Aadil, R.M.; Zeng, X.-A.; Zhang, Z.-H.; Wang, M.-S.; Han, Z.; Jing, H.; Jabbar, S. Thermosonication: A potential technique that influences the quality of grapefruit juice. International Journal of Food Science & Technology 2015, 50, 1275-1282, https://doi.org/10.1111/ijfs.12766.
- Luterotti, S.; Marković, K.; Franko, M.; Bicanic, D.; Madžgalj, A.; Kljak, K. Comparison of spectrophotometric and hplc methods for determination of carotenoids in foods. Food Chem. 2013, 140, 390-397, https://doi.org/10.1016/j.foodchem.2013.02.003.
- Costa, M.G.M.; Fonteles, T.V.; de Jesus, A.L.T.; Almeida, F.D.L.; de Miranda, M.R.A.; Fernandes, F.A.N.; Rodrigues, S. High-intensity ultrasound processing of pineapple juice. Food Bioprocess Technol.2013, 6, 997-1006, https://doi.org/10.1007/s11947-011-0746-9.
- Wang, J.; Wang, J.; Ye, J.; Vanga, S.K.; Raghavan, V. Influence of high-intensity ultrasound on bioactive compounds of strawberry juice: Profiles of ascorbic acid, phenolics, antioxidant activity and microstructure. Food Control 2019, 96, 128-136, https://doi.org/10.1016/j.foodcont.2018.09.007.
- Wang, J.; Vanga, S.K.; Raghavan, V. High-intensity ultrasound processing of kiwifruit juice: Effects on the ascorbic acid, total phenolics, flavonoids and antioxidant capacity. LWT 2019, 107, 299-307, https://doi.org/10.1016/j.lwt.2019.03.024.

Round 2
Reviewer 2 Report
Thanks for the revised manuscript.
Reviewer 3 Report
/